# Numerical and Experimental Assessment of a Novel Anchored for Intramedullary Telescopic Nails Used in Osteogenesis Imperfecta Fractures

**Luis Antonio Aguilar-Pérez** [1]**, José Israel Sánchez-Cruz** [2]**, Juan Alejandro Flores-Campos** [3] **and Christopher René Torres-SanMiguel** [2],*

1   Instituto Politécnico Nacional, Unidad Profesional Interdisciplinaria en Energía y Movilidad, Unidad Zacatenco, 07738 Ciudad de México, Mexico; laguilarp@ipn.mx
2   Instituto Politécnico Nacional, Escuela Superior de Ingeniería Mecánica y Eléctrica, Sección de Estudios de Posgrado e Investigación, Unidad Zacatenco, 07738 Ciudad de México, Mexico; jsanchezc1107@alumno.ipn.mx
3   Instituto Politécnico Nacional, Unidad Profesional Interdisciplinaria en Ingeniería y Tecnologías Avanzadas, 07340 Ciudad de México, Mexico; jaflores@ipn.mx
*   Correspondence: ctorress@ipn.mx

**Featured Application: The problem caused by distal part disengagement is solved by implementing clamping through the yoke in the contact areas between the implant and the bone.**

**Abstract:** Osteogenesis Imperfecta (IO) is a bone disease mainly characterized by the low bone density that produces common fractures in children around 0–7 years. The use of metal implants is a typical treatment of this disease. The intramedullary telescopic nail (ITN) was inspired by the progressive growth in the long bones such as the femur or humerus during children's aging. This work shows an experimental assessment of the ITN's, focusing on their fixation; the proposed improvements in the design of the intramedullary nail studied include the separation of the element into two parts for telescopic enlargement, minimal invasive fixation through the distal anchorage, and the double auto-drilled end for fixation on the distal and proximal section of the bone. The samples were manufactured in 316 L steel and mounted on specialized jaws to replicate the implants' boundary conditions. The experimental test was repeated three times to report the intramedullary telescopic nail's behavior at three lengths. The results show that the device supports only 79.06 N when not at extension length. However, if the device is extended 150% it will support 46.87 N which suggests that intramedullary telescopic nails can only increase by 25% of their original length before they fail.

**Keywords:** biomechanics; osteogenesis imperfecta; intramedullary telescopic nails; prosthesis; testbed

## 1. Introduction

Osteogenesis Imperfecta (IO) stands out from other bone diseases due to its high rarity and clinical severity, typically detected in newborns and child patients. It is characterized by low bone mass, causing several deformations due to body weight. Additionally, long bones like the femur and tibia, among others, become more fragile and susceptible to bone fracture [1]. This has led to treatments that include wearing external elements on the body to decrease possible injuries due to considerable low bone stiffness [2]. Bone-plates, external nails, flexible intramedullary devices, hydromechanics implants, and electromechanic devices treat fractures in fragile bone. In general, all these devices increase bone stiffness by attaching elements directly onto the bone. The main trouble with this solution is due to the fragility of the IO bones. The bones' low density in IO patients decreases the implantation area, causing the implant to slip out of place. In addition, IO children increase in size by almost 2 cm per year and need to replace their implants constantly. For this reason, it is necessary to attach medical devices with minimal invasion and high bone fixation.

One practical solution is the use of intramedullary implants. The procedure described for this kind of implant includes putting metallic elements onto the bone's medullary space.

State-of-the-art intramedullary implants are mainly classified into three types: rigid implants, flexible implants, and telescopic implants. Rigid implants were first used by Küntscher [3]. These devices are characterized by plates or rods fixed to the fractured bone using completely steel nails. Their main advantage is the immediate intervention for the patient. Unfortunately, this type of element cannot be placed in patients with severe bone deformities or those with multiple fractures, since their fixation depends on the patient's available bone area. In anticipation of this in the late 1990s, the Rush brothers proposed the use of flexible nails. The main improvement was that these implants are put in pairs inside the medullary cavity, increasing the implant's adaptability to the bone shape [4]. The patient's operation is less invasive and allows for the combination of this technique with other fracture fixation methods, significantly improving the patient's recovery. The trouble with this method is that it prevents bone growth when used on patients too young. In other words, if the implant remains well-fixed to the ends of the bone, the growth in the patient's bone will be limited to the original size of the implant, causing the need for continuous operations to adapt to the bone growth. To solve this inconvenience, intramedullary telescopic nails (ITN) were developed [5]. These devices are mainly characterized by having an element that dynamically increases the original implant's length by dividing the main element's size into two parts by some types of active or passive mechanisms [6]. The most recent models, such as Bailey and Dubow [7], Sheffield [8], or the Fassier–Duval [9], differ in the way in which each one is anchored to the bone. The main disadvantage of using this kind of device in patients with IO is the small area to anchor the device to enable proper functioning. Auxiliary systems could be used to fix this type of device [10,11]. Another way is using external blocks, although they lack aesthetics and efficiency by remaining for long periods outside the stabilized limb [12]. Thus, reference [13] points out that, until 1999, intramedullary fixation evolution has been lack. The main design parameters have been the cross-transversal section and the locking method to the bone.

By combining the previously mentioned models' results a new model of the intramedullary telescopic nail, with minimal external anchorage on the distal section and screw fixation on the proximal section, was developed. In addition, the intramedullary nail is divided into two parts, allowing them to be extended almost 2/3 of their original length. One of the main methods to fix the device to the bone is locking nails, which are placed orthogonal to the intramedullary nail, crossing the bone through its entire cross-section. On the proposed device, the method used to fix the ITN is an orthogonal anchorage drilled into the bone perpendicular to the male nail. This method is proposed to check three functional characteristics: the maximum deflection allowed by the intramedullary telescopic nail, the maximum payload before the male nail clamped by the locker fails, and the minimum payload necessary to uncouple the entire element. Finite element analysis was performed to obtain a span of expected payload and the entire element's possible behavior. As is shown in reference [14,15], this methodology can be used to improve the design and manufacture of novel implants without being invasive to the patient. By using these results, this work adapted the procedure described by the standard ASTM F1264-16a. This regulation only provides basic geometrical definitions, dimensions, and manufacture recommendations for intramedullary fixation devices. Three mechanical tests were carried out by adapting the regulation ISO 15142-1:2003 that specifies requirements for the manufacture of metallic medical devices used as intramedullary nails to stabilize long bones. The ITN was evaluated by varying the nail's length of the implant's usability by a child and their aging. Another two samples were manufactured to test tension and compression to directly evaluate the locker's behavior.

## 2. Materials and Methods

The device is composed of four elements (Figure 1A. The first part is the thinnest of all, named a male nail (E1). This part is used to guide the second part, known as a female nail (E2). The male nail is anchored distal to the bone, and the female nail is anchored proximal to the bone by screwing them into the bone walls. The male nail is also clamped by the locker (E3) and fixed using one oppressor pin (E4). The locker is drilled into the bone perpendicular to the male nail. By adapting the regulation ASTM F1264-16a, the payload scheme for testing is shown in Figure 1B.

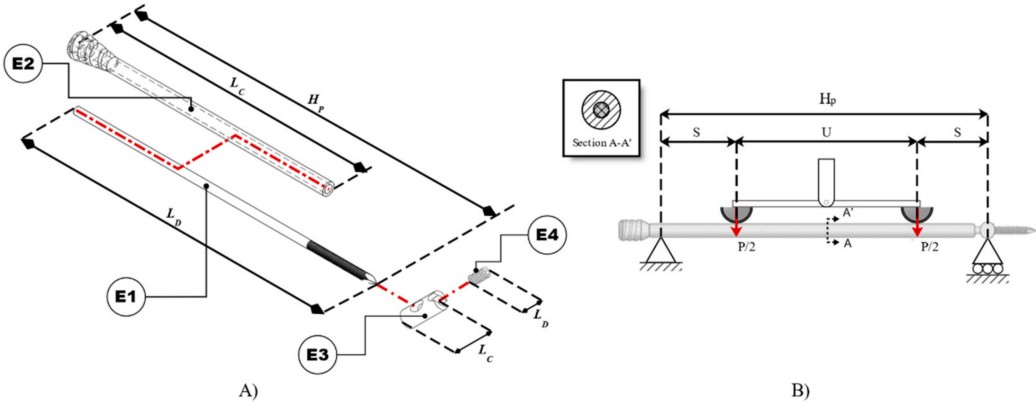

**Figure 1.** Border diagram of the telescopic intramedullary implant; (**A**) ITN parts. (**B**) Standard ASTM F1264-16a.

The HP variable indicates the distance between each nail support. In addition, the LC variable represents the female nail length, and LD represents the male nail length. The load applied on the nail is the variable P. This load is applied equidistantly to the support ends an S variable distance. The separation between each applied load is represented by U. The different diameters of every part of the device are pointed by $f_a$ for the locker diameter, $f_P$ for the oppressor pin, and $f_{H1}$ and $f_{H2}$ represent the outer and inner diameter of the female nail. The variable $f_M$ represents the diameter of the male nail. The entire element's deformation equation for the proposed payload scheme shown in Figure 2 is described by

$$\delta = (P/24E)\,(3Hp_i^2 - 4S_i^2) \tag{1}$$

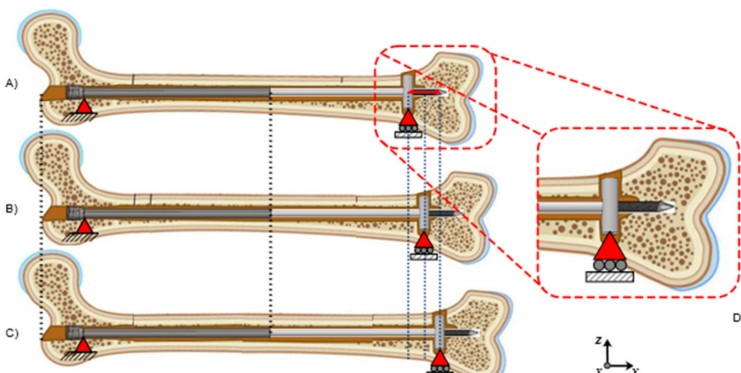

**Figure 2.** Coronal femur view with the telescopic intramedullary implant with double distal anchorage. (**A**) Diagram of non-expanded device. (**B**) Diagram of 2 cm expanded nail. (**C**) Completely extended nail. (**D**) Close view of the anchorage of the male nail.

The $S_i$ and $HP_i$ are the length-dependent extension of the nail. This variable adopts three positions, represented by $HP_0$ (no extension of the intramedullary telescopic nail), $HP_1$ (representing a 30% extension of the intramedullary telescopic nail), and $HP_2$ (representing

a 50% of the extension length of the intramedullary telescopic nail). The described method is applied to evaluate the entire element's length when it is recently implanted and when it needs to be by a maximum of 30% of its original length. Finally, the element is tested when the nail is extended to 50% of its original length. These tests propose to evaluate how much time/length proportion could be possible using the implant for a child before it fails. Studies involving animals or humans, and other studies that require ethical approval, must list the authority that provided approval and the corresponding ethical approval code.

The modulus of elasticity is constant, and inertia moment I is from $4\pi$ and is obtained from Equation (2), where the geometries of the intramedullary telescopic nail for bone growth are considered.

$$I = \pi \left( \left( f_{H1}/2 \right)^4 - \left( f_{H2}/2 \right)^4 + \left( f_M/2 \right)^4 \right) /4 \tag{2}$$

The proposed device will be implanted in the femur of children with IO. In this way, the model is anchored proximally and distally. Additionally, it extends dynamically by separating the nail into two parts, allowing them to adapt their length simultaneously as the children's femur size increases. In addition, the nail is fixed by an external locker drilled into the bone perpendicular to the medulla cavity. For this reason, the boundary conditions are shown in Figure 2. Figure 2A shows the proposed anchorage in the femur. Figure 2B compares the elongation of the femur when the children have grown by around 2 cm. Additionally, Figure 2C presents an elongation of 5 cm. Figure 2D presents a closer view of the proposed orthogonal anchorage of the male nail.

### 2.1. Four-Bending Testing

The five ITN samples were manufactured in stainless steel of 316 L surgical grade, with the capacity to elongate from 120 mm to 200 mm. The surface was not polished, to avoid reflections during the tests. No thermal, chemical, prestressing, or relaxation cycle treatment was carried out after the piece's manufacture. A random light paint spotting was carried out on the device's male and female nails (Figure 3). This preparation was proposed to measure the displacements of the device using photogrammetry. A homogeneously matte white paint base was applied over the sample's entire open area (Figure 3A). Each sample was then mottled using black aerosol paint, approximately 50 cm from the horizontal specimen. The final mottled sample is shown in Figure 3B.

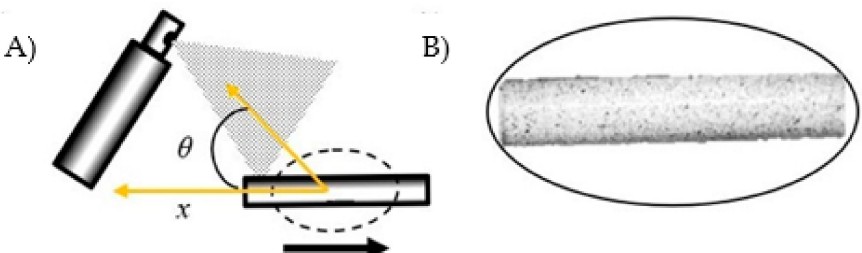

**Figure 3.** Photogrammetry sample; (**A**) Mottled of the sample by using black aerosol. (**B**) Sample preparation.

The intramedullary telescopic nail with distal anchorage was evaluated in the National Polytechnic Institute's materials testing laboratory in Mexico City. The machine used was a Shimadzu universal testing model AG-IS 100 KN. The tests were performed by applying incremental loads with a speed of 1 mm/s until the system ceased to present a functionality. A customized jaw system was designed to perform the 4-point flex. The custom jaw system is shown in Figure 4.

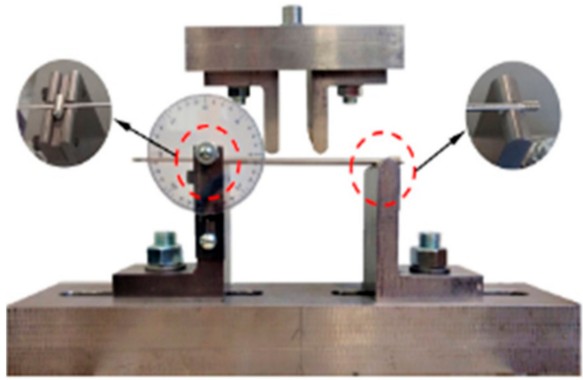

**Figure 4.** Intramedullary telescopic nail testbed under 4-point flexion test.

*2.2. Numerical Analysis of Four-Bending Testing*

The intramedullary telescopic nail with double distal anchorage was evaluated by performing a numerical simulation using the Finite Element Method [16]. The contact conditions were established for the interaction between the female and male nails, as well as the jaws (Figure 5A). The FEBIO software was used and the contact defined was "sticky". The Young's module used for the simulation was 200 GPa and the Poisson's rate was set at 0.3. The separation before contact was 0.1 mm and the load was applied slowly ($5 \times 10^{-4}$ s). The payload was determined using Equation one. For the first case study, the implant was not extended and it was applied at a displacement of 0.1714 mm to deform the element (Figure 5B). Subsequently, the intramedullary telescopic nail was extended by one-third of its maximum length and 1.6107 mm of displacement was applied (Figure 5C). In the third case study, two-thirds of its length was extended and a displacement of 4.3553 mm was applied (Figure 5D).

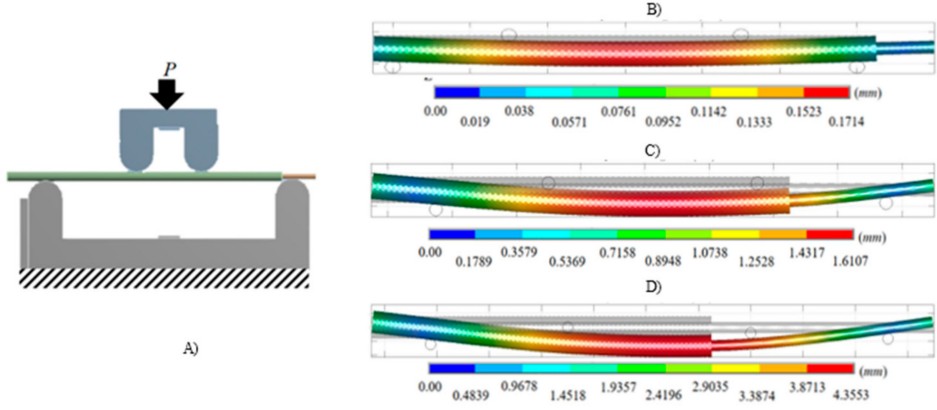

**Figure 5.** (**A**) Loads applied to the telescopic intramedullary implant (**B**) Case study 1. (**C**) Case study 2. (**D**) Case study 3.

## 3. Experimental Results

The results obtained from the experimental four-point bending tests that were carried out are presented in Figure 6. In case 1, the cross-section remains constant for all sections; in case 2, the cross-section is constant for two of the three sections. In case 3, the cross-section is entirely different for each section. Thus, the results of the three tests are listed next: for case 1, the reached stress value was closed to 251.6496 MPa with a deflection of 0.843 mm at approximately −20 mm from the implant center, while in case 2, the stress values were 254.6313 MPa with a deflection of 1.7180 mm at approximately −26.66 mm from the center of the implant, and for case 3, the maximum stress value was 248.6541 MPa with a total deflection of 4.929 mm. Figure 6 shows the complete results reported by the software of the universal testing machine. Figure 6C presents a close view of linear

behavior. This approach makes it possible to visualize that the first two cases support a greater load capacity within the elastic limit of the material around 250 MPa. The third case suffers a total deformation of the element, even before it reaches the elastic limit. The loads recorded at the limit of the linear behavior are approximately 79.058 N for case 1 (red line with circles), 60 N for case 2 (blue line with circles), and 46.875 N for case 3 (green line with dashes).

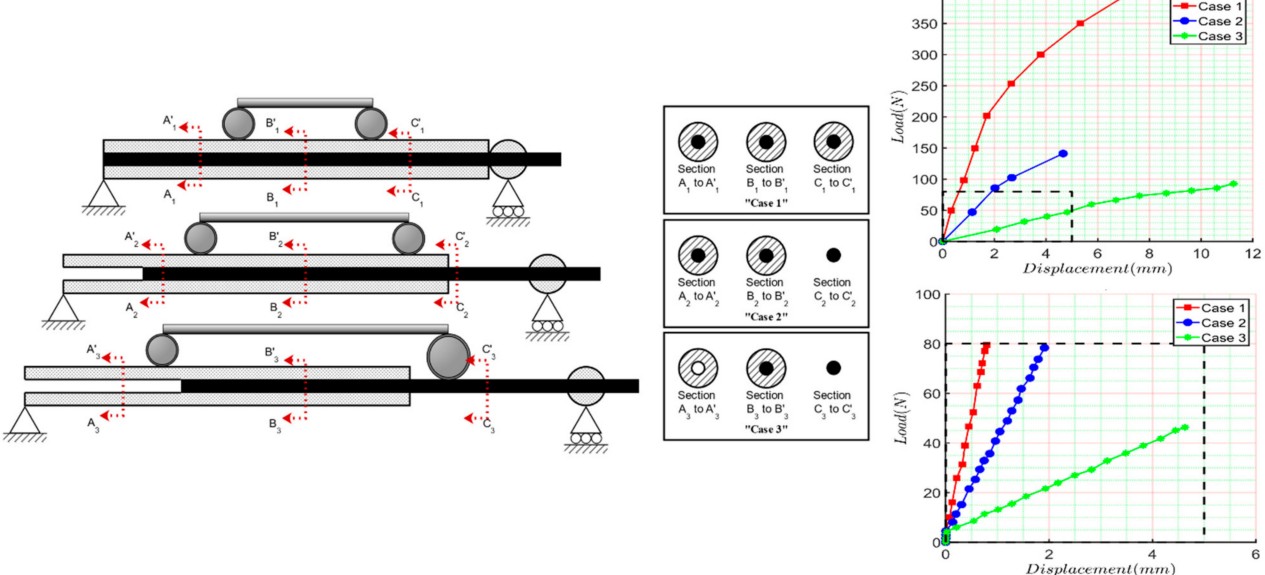

**Figure 6.** (**A**) Cross-section comparative for the four bending tests. (**B**) Total behavior of the experimental test. (**C**) Data acquired from the experimental test.

All the tests were filmed to analyze the data after the run by using the GOM-Inspect software. Figure 7A shows the deformations reached on the male sample for case 1; the maximum deflection obtained was 0.80 mm (Figure 7B). The computer program indicates a maximum deflection of 1.72 mm (Figure 7C). Finally, for the third case, a maximum deflection of 4.92 mm was obtained according to the analysis (Figure 7D).

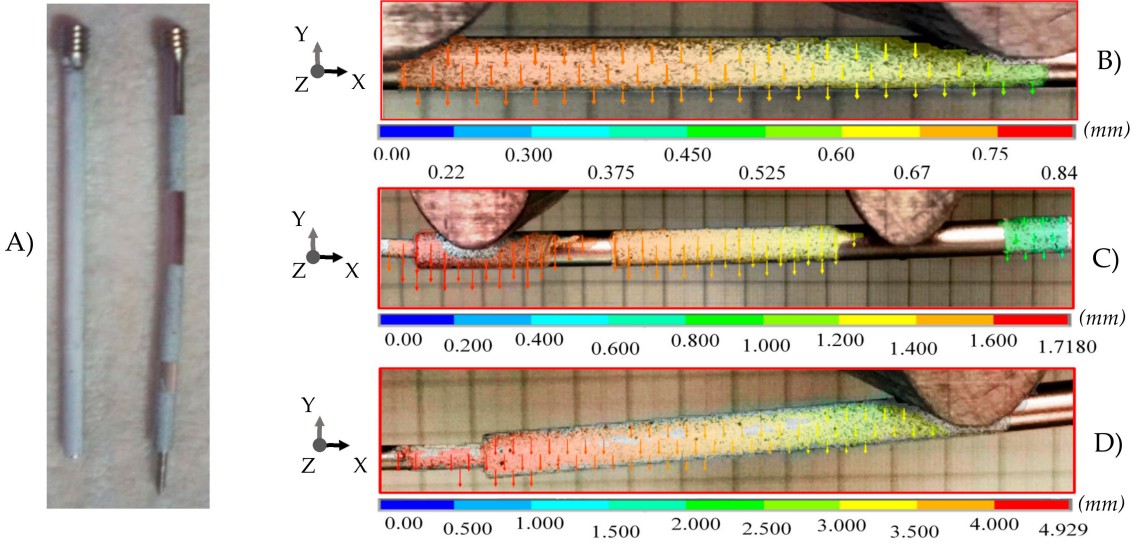

**Figure 7.** Results of experimental tests (**A**) Samples after deformation. (**B**) Correlation of digital images for case 1. (**C**) Case 2. (**D**) Case 3.

## 4. Numerical Results

Several simulations were run considering different mesh conditions and types of contact interaction (Figure 8). The first contact was performed on hexahedral mesh (Figure 8B), and the contact interaction was performed as a sticky interface. A sticky interface is similar to a tied interface, except that it allows an initial separation of the surfaces, and it breaks the "tie" condition after standard traction. The second contact was performed on a tetrahedral mesh (Figure 8B), and the contact interaction was performed as a sliding elastic interface. The sliding contact interface uses facet-on-facet contact but differs in the linearization of the contact forces. Thus, this contact definition considers a frictionless contact interface. The contact interfaces are shown in Figure 8A.

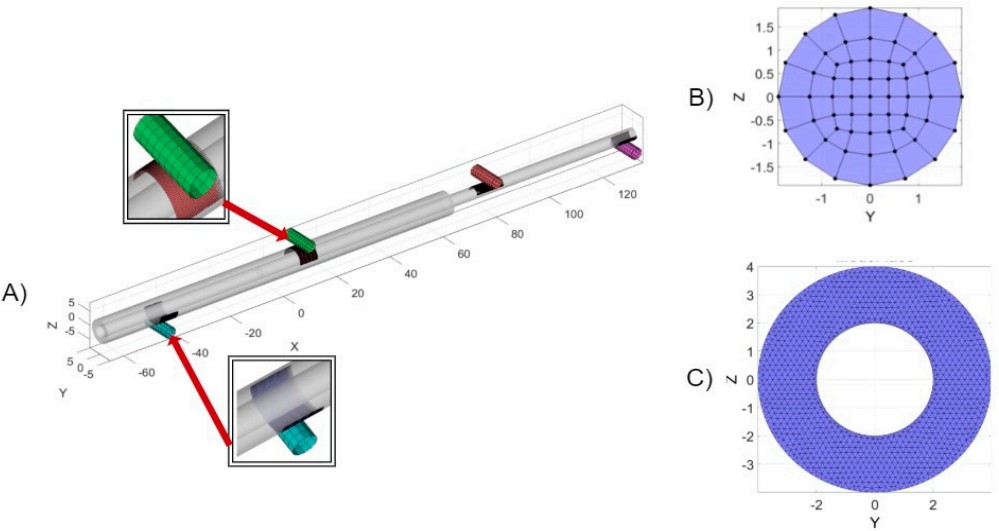

**Figure 8.** Mesh conditions for experimental contact comparative; (**A**) Definition of region interactions. (**B**) Hexahedral 8 nodes linear mesh. (**C**) Tetrahedral 4 nodes linear mesh.

The comparison of the number of elements versus the time elapsed is shown in Figure 9A. The comparative of the number of elements versus the Von Mises stresses is shown in Figure 9B. The comparative number of elements versus the displacement is shown in Figure 9C.

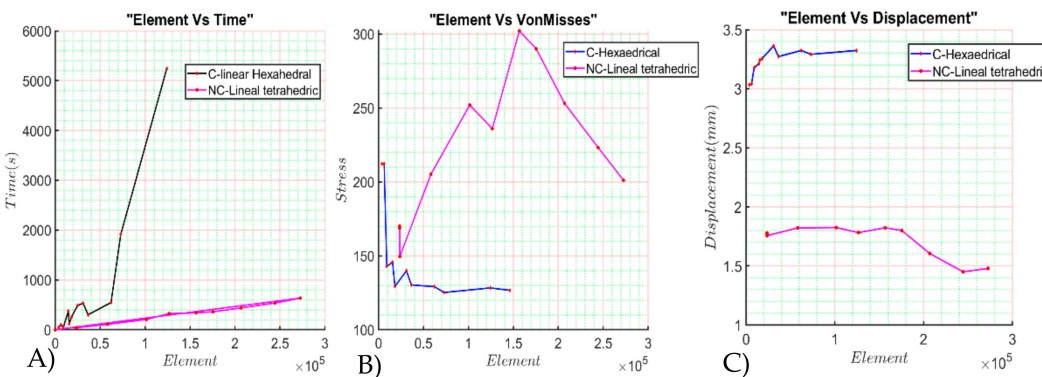

**Figure 9.** Mesh convergence analysis; (**A**)Elements versus time. (**B**) Elements versus Von Mises. (**C**) Elements Versus deformation.

The following table summarizes the computed results for hexahedral and tetrahedral elements with contact and no contact. The number of elements is modified through geometrical parameters, like the dimension, width, or depth of the element.

By looking at the data shown in Table 1, it can be observed that values 7 and 11 are equal for several elements, but the computed data are quite different. This behavior is similar to reported values 8 and 12. The main reason for the duplicated number of elements is because they are parallel to the large axis. The results shows that the elements' numbers of depth is less significant than the number of radial elements of the simulation.

**Table 1.** Computed values for different mesh.

| Test | Elements | VM Value | $D_z$ Mean | Time | Type of Element | Contact Condition |
|------|----------|----------|--------|------|-----------------|-------------------|
| 1 | 3648.00 | 212.23 | 3.0329 | 44.81 | | |
| 2 | 6208.00 | 212.18 | 3.0415 | 97.15 | | |
| 3 | 9120.00 | 142.86 | 3.1781 | 55.25 | | |
| 4 | 15,520.00 | 145.73 | 3.242 | 119.36 | | |
| 5 | 18,240.00 | 129.32 | 3.2561 | 261.61 | | |
| 6 | 31,040.00 | 139.91 | 3.3636 | 534.08 | | |
| 7 | 36,480.00 | 126.49 | 3.2719 | 300.91 | Hexahedrical linear | Sticky contact |
| 8 | 62,080.00 | 129.19 | 3.3241 | 546.87 | | |
| 9 | 14,592.00 | 752.07 | 3.0163 | 374.70 | | |
| 10 | 24,832.00 | 784.86 | 2.817 | 486.77 | | |
| 11 | 36,480.00 | 130.28 | 3.2402 | 453.33 | | |
| 12 | 62,080.00 | 142.62 | 3.2824 | 1364.70 | | |
| 13 | 72,960.00 | 125.19 | 3.2912 | 1918.10 | | |
| 14 | 124,160.00 | 128.30 | 3.3238 | 5247.00 | | |
| 15 | 145,920.00 | 126.73 | 3.1404 | 960.00 | | |
| 16 | 24,640.00 | 332.84 | 2.6439 | 42.82 | Tetrahedral linear | Sticky contact |
| 17 | 49,280.00 | 432.53 | 2.8139 | 93.54 | | |
| 18 | 98,560.00 | 2184.20 | 4.1438 | 273.23 | | |
| 19 | 23,499.00 | 168.7178 | 1.7733 | 39 | | |
| 20 | 23,538.00 | 170.1108 | 1.7787 | 40.6458 | | |
| 21 | 23,646.00 | 149.4692 | 1.7566 | 41.33 | | |
| 22 | 58,274.00 | 205.1384 | 1.8222 | 109.2592 | | |
| 23 | 101,488.00 | 251.9337 | 1.8249 | 206.84 | Tetrahedral linear | Sliding elastic |
| 24 | 126,662.00 | 235.9668 | 1.7822 | 324.3423 | | |
| 25 | 147,773.00 | 209.373 | 1.6077 | 322.0232 | | |
| 26 | 206,997.00 | 253.1276 | 1.6049 | 437.8151 | | |
| 27 | 244,466.00 | 223.1868 | 1.4508 | 537.00791 | | |
| 28 | 272,644.00 | 201.0864 | 1.4787 | 635.524 | | |
| 29 | 339,645.00 | 439.0501 | 1.8002 | 762.511 | | |

## 5. Discussion

Our study provides additional support for future analysis for the implantation of intramedullary telescopic nails. The study relies on two main aspects, the dynamic variation in the length as the implant remains in the body when children are grown, and the distal anchorage reinforcement achieved by the orthogonal locker. The load versus displacement graph shown in Figure 6 matches with the statement of the regulation ASTM F1264-16a that points out the maximum separation of the loads and the supports must be under 1/3 of the total length. However, further tests must be carried out varying the element's length to include the two parts of the intramedullary nail's behavior. The importance of these studies is shown in the deflection diagram in Figure 10.

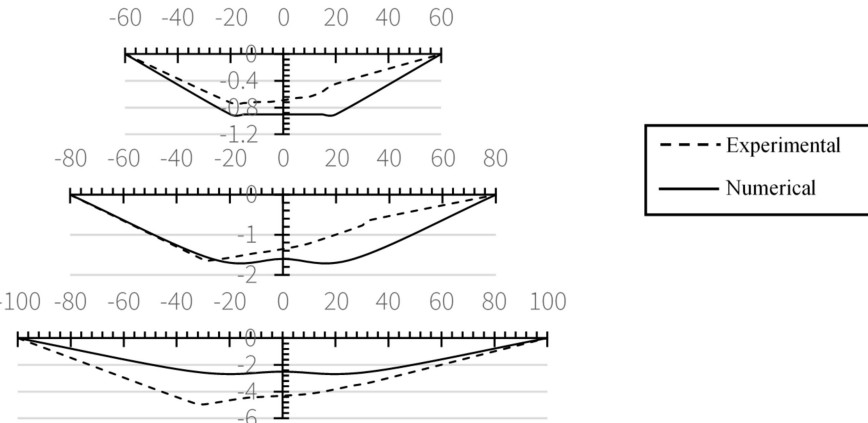

**Figure 10.** Graphical comparative of the maximum displacement of the intramedullary telescopic nail.

The deflection diagram identifies the most probable region where the implant fails due to the increased length of the implant. On each test, the maximum deflection matches the transition in the male nail and the female nail. For case 1, the numerical simulation reaches a stress value of 251.6496 MPa at approximately $-20$ mm from the implant center, while in case 2 and case 3 this value moves to $-26.66$ mm from the center of the implant and approximately $-33.33$ mm, respectively, matching the values obtained by the experimental tests. The small cross-section of this element modifies the implant's expected performance, and the probability of suffering a fracture is increased. In order to improve this behavior, we propose a change in the material that manufactures the male nail increasing the strength of the implant. Another way the intramedullary telescopic nail with distal anchorage is loosened is due to the oppressor pin on the male nail. The distal anchorage system fails when a load of 216.4063 N is applied. This load's value is close to $-858$ N on the Z-axis of the femur bone [17,18] and is reachable by dynamic loads like falls from medium heights. In this way, if the patient does not resist the fall the implant will likely suffer a plastic deformation and, in consequence, the remotion of the implant.

In order to validate the experimental results, we have carried out several numerical tests (around 30 tests), varying the type of element, the density of the mesh, and the contact properties. We present a comparative of the numerical results in Figure 11. For all the numerical tests reported, we refer to the case 3 for the geometrical positions for simplicity.

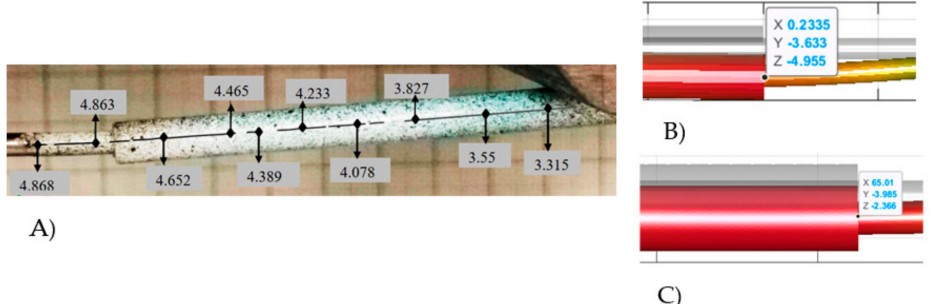

**Figure 11.** Comparative of numerical displacement. (**A**) Experimental test. (**B**) Numerical solution considering contact conditions. (**C**) Numerical conditions considering no contact conditions.

By comparing all the numerical results, we conclude that male-to-female interaction causes a deviation in the z-direction and y-axis. This deviation was not considered or not evident in the experimental test due to the distal and proximal anchorage but raised some doubts as to how this deviation could modify the stability of the element if the tolerance of the interface female–male is too high. We need to improve further testing in order to validate this consideration.

## 6. Conclusions

In the present work, an experimental methodology has been presented to evaluate the ability of intramedullary telescopic nails with distal anchorage to stabilize fractures in patients with low bone mass like those with Osteogenesis Imperfecta. The proposed methodology follows the procedure described by the regulation ASTM F1264-16a to evaluate the strength of the intramedullary nails. In addition, compression and tension tests were developed to evaluate the clamping of an orthogonal locker. Of the complete study, three samples were subjected to a four-point bending test and the compression and tension of two more samples were tested. The three first samples' responses showed that the ITN device works exceptionally well under flexion loads until one-third of the device's total extension is reached. After that length extension, the nail only supports half of the total admissible original load. In addition, the importance of this study shows that the ductile area of the device moves in the same way as when the female–male joint starts to be extended. In future study improvements, the male nail will be manufactured and tested in strengthened materials to decrease this ductile behavior. Additionally, the tension and compression tests show necessary behavior not considered in the initial device design. Further examinations of the joint oppressor pin's backlash to the intramedullary male nail are necessary to decrease the bone's unfixing. It is imperative to improve, within the anchorage system, a mechanism that guarantees the fixation caused by the oppressor pin on the male nail because the operation of this part of the device represents almost 50% of the complete implant operation.

## 7. Patents

Patent MX/u/2019/000072 results from work reported in this manuscript.

**Author Contributions:** Conceptualization, C.R.T.-S. and L.A.A.-P.; methodology, C.R.T.-S.; software, J.I.S.-C.; validation, L.A.A.-P. and J.I.S.-C.; formal analysis, J.I.S.-C.; investigation, J.A.F.-C.; resources, C.R.T.-S.; data curation, J.A.F.-C.; writing—original draft preparation, J.I.S.-C.; writing—review and editing, C.R.T.-S.; visualization, J.A.F.-C.; supervision, C.R.T.-S.; project administration, J.A.F.-C.; funding acquisition, C.R.T.-S. All authors have read and agreed to the published version of the manuscript.

**Funding:** This research received no external funding.

**Institutional Review Board Statement:** Not applicable.

**Informed Consent Statement:** Not applicable.

**Data Availability Statement:** Not applicable.

**Acknowledgments:** Government of Mexico by the National Council of Science and Technology (CONACYT) and the Instituto Politécnico Nacional. Authors acknowledge partial support from project 20210282 and EDI grant, all provided by SIP/IPN.

**Conflicts of Interest:** The authors declare no conflict of interest.

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
