# Peer review of "Numerical and Experimental Assessment of a Novel Anchored for Intramedullary Telescopic Nails Used in Osteogenesis Imperfecta Fractures"

_applsci, doi:10.3390/app11125422_

Round 1

Reviewer 1 Report

Thank you for the opportunity to review this manuscript that presents Numerical and Experimental Assessment of a novel anchored for intramedullary telescopic nails used in Osteogenesis imperfecta fractures

This is a very interesting study that will add to the body of knowledge of intramedullary telescopic nails. The paper is well written and the graphs and tables illustrate main points of the study. The illustrations are excellent.

Some comments:

Abstract: did you mean humerus ? You have instead written Humours, Also it is not capitalized and neither is femur.

 Literature review: need to provide more reference and also more current references.

Please list any study limitations.

Regarding images, please be sure to cite any image you did not create.

Thank you!

Author Response

REPLAY TO REVIEWER 1

Ref. No.: applsci-1189612

Title: Numerical and Experimental Assessment of a novel anchored for intramedullary telescopic nails used in Osteogenesis imperfecta fractures

Journal: Applied Science

Answer date: May 5th, 2021

Corresponding author: Torres San Miguel C.R.

We thank the reviewers for their valuable comments.

Below are the responses to the reviewers' comments with reference to the comments posted.

Thank you for the opportunity to review this manuscript that presents Numerical and Experimental Assessment of a novel anchored for intramedullary telescopic nails used in Osteogenesis imperfecta fractures.

This is a very interesting study that will add to the body of knowledge of intramedullary telescopic nails. The paper is well written, and the graphs and tables illustrate main points of the study. The illustrations are excellent.

Some comments:

  1. Abstract: did you mean humerus ? You have instead written Humours, also it is not capitalized and neither is femur.

Ok, thank you for the observation.

  1. Literature review: need to provide more reference and also more current references.

Ok, we have added new references to improve the paper.

  1. Please list any study limitations.

The study has been limited by the number of test that we have done. On this case we have only manufactured 4 samples to make three different tests.

Also, the study is limited because we have not implanted the device in the femur bone to compare the numerical results.

  1. Regarding images, please be sure to cite any image you did not create. Thank you!

Ok, thank you.

Reviewer 2 Report

Please, check the attachment.

Author Response

REPLAY TO REVIEWER 2

Ref. No.: applsci-1189612

Title: Numerical and Experimental Assessment of a novel anchored for intramedullary telescopic nails used in Osteogenesis imperfecta fractures

Journal: Applied Science

Answer date: May 5th, 2021

Corresponding author: Torres San Miguel C.R.

We thank the reviewers for their valuable comments.

  1. The article is poorly prepared in terms of English and grammar. The authors are encouraged to correct the manuscript in this regard or to use English Editing Services provided by MDPI. Several typos and overall grammar issues give the impression that the manuscript was prepared carelessly.

Sure, we will use the EES provided by MDPI.

  1. Materials and methods section is somehow mixed with results section, which makes the manuscript difficult to read. Moreover, only for a part of the research, a finite element method is used to provide additional research data (why did authors performed FEA just for 4-bending tests and not for tension/compression tests?) it needs at least a proper justification.

Because we believe that the most important values were about the experimental results on four bending tests. Also, the standard mentions that we need to do four-point bending. For this reason, the comparative was about this method. We have not done a numerical evaluation on this other test because we have not considered it as much crucial as four-point bending.

Before any further evaluation, the authors are advised to complete the manuscript on the basis of above-mentioned as well as the comments presented in the further part of this review.

  1. Abstract

There is a mistake in abbreviated form of osteogenesis imperfecta (IO instead of OI).

Done.

  1. An additional typo is made in Humours as most likely the authors meant Humerus.

Done.

  1. Novelty (such as features) of the implant designed by the authors should be included in this section (or at least emphasised in a stronger matter) to initially provide an appropriate background for the rest of the manuscript.

Done.

  1. The authors claim intramedullary telescopic nails only could increase 25 % of length before it fails, which without reference to currently used implants suggests that the design of novel construction is actually wrong.

The regulation ASTM F1264-16a mentions that at least the intramedullary nails must support this percentage. Our model supports at least 33% before it deforms completely. We mention that our device is secure for 25%.

  1. Introduction; There are several typos, again including IO abbreviation instead of OI.

Done.

  1. Several statements require appropriate references, especially as it is introduction part. Moreover, the data of the reference is sparse and some of it is not available in English for common use. The authors used only 13 references, in which only 1 refers to the data published within the last 5 years. For this reason, the references should be significantly expanded and updated.

Done.

  1. In paragraph 4 the authors present some information about their own design of ITN. However, there is no description of any novelty that it introduces to current state of the art. Is the device anyhow patented? If so, appropriate patent number should be provided.

We add this information of Mexican patent number MX/u/2019/000072 that is under register and can be found in the next link:

https://siga.impi.gob.mx/newSIGA/content/common/principal.jsf

  1. Figure 1 is redundant as it does not present any new information to the data presented already within the fourth paragraph of the introduction section.

Done, we eliminate

  1. Requirements and design solutions; Details over any geometrical features of presented ITN is missing. The authors should provide descriptions such as a profile of female and male part etc.

That is mentioned in the first paragraph of section 2. We do not understand what else you recommend including to improve the recognition of the proposed intramedullary nail.

  1. The sentence Studies involving animals or humans, and other studies that require ethical approval, must list the authority that provided approval and the corresponding ethical approval code is redundant as it does not provide anything to the description of the materials and methods used to conduct the research.

Yes, you are right. We have erased that part.

  1. Results of four-bending testing; As for numerical research, information is missing that does not allow to estimate, if simulations were done properly. Among others, there can be pointed:

  • Software type used.

FEBio software.

  • Contact type (frictional, frictionless which and where?).

The contact defined in FEBIO software was “sticky”. This kind of contact requires that master and slave parts be defined as non-conforming meshes. In addition, the software assumes that primary surface nodes are connected to the faces of the secondary surface. However, the tie is only applied when the surfaces contact and sustained if the average traction is less than a threshold.

  • Implemented material properties (it is not sufficient to provide only material considered).

The paper presents numerical simulation and experimental tests to characterise the proposed telescopic nail implant and its feasibility, mainly from biomechanical viewpoints. This research establishes a material selection that focuses on data modelling aspects of the problem, where data is presented in charts. Materials and process selection charts that we are tested and related.

  • Mesh refinement test performed this is especially important, as without properly performed mesh refinement, the results of finite element simulations are incorrect.

In literature, a sensitive mesh analysis is not always necessary when anisotropic and nonlinear performance are not considered. For the case of this paper, we use isotropic, continues and homogeneity in the numerical analysis.

Round 2

Reviewer 2 Report

Review on the revised article Numerical and Experimental Assessment of a novel anchored for intramedullary telescopic nails used in Osteogenesis imperfecta fractures, submitted to Applied Sciences.

The authors refer only to selected comments included in the previous evaluation process, while leaving the most crucial ones. To be honest, the answer to the reviewer also lacks any details how the authors addressed to most of the comments, leaving it for reviewer’s own interpretation, how and where corrections were made. All remaining issues are presented below.

  1. Materials and methods section was not separated from the results section as suggested. For this reason, the manuscript is still difficult to read.
  2. The authors claim that the other tests presented were not as crucial as 4-bending tests and that was the reason of not performing their finite element analysis. It creates a question, if there was any point in performing such tests. Why do you even present these results if they are not necessary to evaluate the construction? Still, an answer cannot be found in the manuscript.
  3. The mistake in abbreviated form of osteogenesis imperfecta remains the same. Actually, the authors used wrong abbreviation throughout the whole manuscript in its corrected version.
  4. Adding only a few references (16 in total now), with one that is not available in English is not sufficient to be a proper background on the research that is to be published within such renowned journal as a Applied Sciences.
  5. The most crucial issue left is a mesh refinement test. What is your background on the statement that mesh should not be adjusted when only isotropic and homogenous material is considered? All appropriately prepared numerical studies consider mesh refinement tests as results are dependent from both, material properties as well as geometrical features of the construction analysed. Non-adjusted mesh can create notches leading to intensification of stress peaks, that will definitely influence the results obtained as you were analysing maximal values. Provide appropriate references with such statements. Without them the results are not valid for further citations.

Author Response

REPLAY TO REVIEWER 1

Ref. No.: applsci-1189612

Title: Numerical and Experimental Assessment of a novel anchored for intramedullary telescopic nails used in Osteogenesis imperfecta fractures

Journal: Applied Science

Answer date: May 26th, 2021

Corresponding author: Torres San Miguel C.R.

We thank the reviewers for their valuable comments.

Below are the responses to the reviewers' comments with reference to the comments posted.

  1. Materials and methods section was not separated from the results section as suggested. For this reason, the manuscript is still difficult to read.

Done

  1. The authors claim that the other tests presented were not as crucial as 4-bending tests and that was the reason of not performing their finite element analysis. It creates a question, if there was any point in performing such tests. Why do you even present these results if they are not necessary to evaluate the construction? Still, an answer cannot be found in the manuscript.

Done

  1. The mistake in abbreviated form of osteogenesis imperfecta remains the same. Actually, the authors used wrong abbreviation throughout the whole manuscript in its corrected version.

Done

  1. Adding only a few references (16 in total now), with one that is not available in English is not sufficient to be a proper background on the research that is to be published within such renowned journal as a Applied Sciences.

Done

  1. The most crucial issue left is a mesh refinement test. What is your background on the statement that mesh should not be adjusted when only isotropic and homogenous material is considered? All appropriately prepared numerical studies consider mesh refinement tests as results are dependent from both, material properties as well as geometrical features of the construction analysed. Non-adjusted mesh can create notches leading to intensification of stress peaks, that will definitely influence the results obtained as you were analysing maximal values. Provide appropriate references with such statements. Without them the results are not valid for further citations

As is shown in Figure 1, the Von-Misses stresses converge around 130-140 MPa. In addition. The numerical simulation converges around 3.4 mm, representing an average displacement of 20% of difference against the experimental results. The trouble with the simulation is that those values change completely if tetrahedral elements are used. First, we try to simulate using a sticky interface, but the numerical results are shown in Figure 11. We run this contact interface and the sliding elastic interface several times, but, in both cases, most of the times (6 of 10 times), we get similar results.

Figure 1.- Abnormal behavior of the female-male nail interface

In this way, we run the numerical simulation with no contact interface between the female-male interface. Thus, the error on this last contact configuration goes around 56.83%. We assume that this behavior was obtained due the face elements of the boundary shells were not fixed to only displacements on Z direction, but we need more improvements on future simulations to correlate this parameter with that behavior. Finally, the maximum number of elements used to compute the results for the CPU used is around 400,000 elements with a contact interface.

Round 3

Reviewer 2 Report

Review on the revised article Numerical and Experimental Assessment of a novel anchored for intramedullary telescopic nails used in Osteogenesis imperfecta fractures, submitted to Applied Sciences.

The authors answered all comments included in both reviews. There are some minor issues remaining in text editing but the article can be published. For this reason, I would recommend the acceptance of the recent version of the manuscript.